# Managing Minds at Work: Development of a Digital Line Manager Training Program

**DOI:** 10.3390/ijerph19138006

**Published:** 2022-06-29

**Authors:** Holly Blake, Benjamin Vaughan, Craig Bartle, Jo Yarker, Fehmidah Munir, Steven Marwaha, Guy Daly, Sean Russell, Caroline Meyer, Juliet Hassard, Louise Thomson

**Affiliations:** 1School of Health Sciences, University of Nottingham, Nottingham NG7 2HA, UK; 2NIHR Nottingham Biomedical Research Centre, Nottingham NG7 2UH, UK; 3Institute of Mental Health, Nottinghamshire Healthcare NHS Trust, Nottingham NG3 6AA, UK; ben.vaughan@nottshc.nhs.uk (B.V.); craig.bartle@nottshc.nhs.uk (C.B.); louise.thomson@nottingham.ac.uk (L.T.); 4Birkbeck, University of London, London WC1H 0PD, UK; j.yarker@bbk.ac.uk; 5School of Sport, Exercise & Health Sciences, Loughborough University, Loughborough LE11 3TU, UK; f.munir@lboro.ac.uk; 6Institute for Mental Health, University of Birmingham, Birmingham B15 2TT, UK; s.marwaha@bham.ac.uk; 7Office of the Provost, The British University in Egypt, El Sherouk City 11837, Cairo, Egypt; guy.daly@bue.edu.eg; 8Faculty of Health and Life Sciences, Coventry University, Coventry CV1 5FB, UK; 9Thrive at Work, West Midlands Combined Authority, Birmingham B19 3SD, UK; sean.russell@wmca.org.uk; 10Executive Office, Warwick University, Coventry CV4 7AL, UK; c.meyer@warwick.ac.uk; 11School of Medicine, University of Nottingham Jubilee Campus, Nottingham NG8 1BB, UK; juliet.hassard@nottingham.ac.uk

**Keywords:** digital, mental health, intervention, training, stress, occupational, workplace, workforce, line managers

## Abstract

Mental ill-health is the leading cause of sickness absence, creating a high economic burden. Workplace interventions aimed at supporting employers in the prevention of mental ill-health in the workforce are urgently required. Managing Minds at Work is a digital intervention aimed at supporting line managers in promoting better mental health at work through a preventative approach. This intervention was developed as part of the Mental Health and Productivity Pilot, a wider initiative aimed at supporting employers across the Midlands region of the United Kingdom to improve the future of workplace mental health and wellbeing. The aim of the study is to describe the design and development of the Managing Minds at Work digital training program, prior to feasibility testing. We adopted a collaborative participatory design involving co-design (users as partners) and principles of user-centred design (pilot and usability testing). An agile methodology was used to co-create intervention content with a stakeholder virtual community of practice. Development processes were mapped to core elements of the Medical Research Council (MRC) framework for developing and evaluating complex interventions. The program covers five broad areas: (i) promoting self-care techniques among line managers; (ii) designing work to prevent work-related stress; (iii) management competencies to prevent and reduce stress; (iv) having conversations with employees about mental health; (v) building a psychologically safe work environment. It was considered by stakeholders to be appropriate for any type of organization, irrespective of their size or resources. Pilot and usability testing (*n* = 37 surveys) aligned with the Technology Acceptance Model (TAM) demonstrated that the program was perceived to be useful, relevant, and easy to use by managers across sectors, organization types, and sizes. We identified positive impacts on manager attitudes and behavioral intentions related to preventing mental ill-health and promoting good mental wellbeing at work. The next step is to explore the feasibility and acceptability of Managing Minds at Work with line managers in diverse employment settings.

## 1. Introduction

Mental health problems affect one in six workers each year and are the leading cause of sickness absence, with stress, anxiety, and depression being responsible for approximately half of the working days lost [1,2]. Mental ill-health can lead to unemployment [3], which has broader implications for individuals, employers, and society. The economic burden of mental ill-health is high and likely to increase due to the rising prevalence of mental ill-health due to the COVID-19 pandemic [4], coupled with the impacts of the pandemic on the labor market [5]. A systematic review [6] focusing only on work-related stress (WRS) and including studies from Australia, Canada, Denmark, France, Sweden, Switzerland, the United Kingdom, and the EU-15 suggested that there is a sizable financial burden imposed by WRS on society, ranging from US$221.13 million to $187 billion. Most of the cost of WRS comes from productivity losses—between 70–90%—with the remaining 10–30% coming from health and medical costs [6].

Preventing and managing mental ill-health at work is therefore a priority, not only from a public health perspective, but for organizations and their ‘bottom line’ (net profits). Yet, many employers are unaware of their important role in supporting workers’ mental health [2]. Interventions to mitigate the impact of mental ill-health in the workplace have been typically categorized in ways that distinguish between the stages of prevention and their associated targets of change: primary-level interventions (taking action to eliminate the sources of stress or poor mental health in the workplace), secondary-level interventions (detecting and managing experienced stress or mental health concerns by increasing employees’ awareness, knowledge, skills, and coping resources), or tertiary-level interventions (minimizing the effects of poor mental health at work once they have occurred through the treatment of symptoms and provision of remedial support). A holistic approach incorporating all three types of interventions is needed and advocated as best practice in European guidelines, including the European Foundation for the Improvement of Living and Working Conditions and the European Agency for Safety and Health at Work [7]. However, primary prevention is advocated as being particularly important to ‘flatten the curve’ and avoid a surge in the incidence of mental disorders stemming from the COVID-19 pandemic [8].

Line managers play an important role in the primary prevention of mental ill-health [9]. They can act proactively through the design and management of work tasks, communicating with and managing their employees with respect and clarity, developing a supportive and psychologically safe work environment for their teams, and encouraging people to have conversations about their mental wellbeing at work [10,11,12,13,14,15]. To effectively fulfil this role, managers need to be equipped with the right skills and knowledge, but there is a dearth of rigorously developed guidance to assist with this [16]. Training for line managers has been identified as an important part of the process of supporting employee mental health [12,17]. A systematic review and meta-analysis indicates that training managers in workplace mental health can improve their knowledge, attitudes, and self-reported behavior in supporting employees experiencing mental health problems [18]. Yet, a recent survey showed that line managers are not being given enough support and training to protect the mental health and wellbeing of staff at work, and more than half of organizations (57%) indicated that their organization offers no mental health and wellbeing training and/or support for managerial staff [19]. Where training is offered and shows promise, the development of content and the evaluation of training is often poorly described and/or lacks rigor, hindering its replicability (e.g., [20])

Several studies have examined the effectiveness of line manager training to improve the mental health of employees using trial methodology [17,21,22,23] These have broadly shown that training can lead to improvements in line managers’ knowledge of mental health [22], communication about mental health and related resources [17], confidence in and self-efficacy for creating a mentally healthy workplace [21,22], and lowered work-related sickness absences of employees [23]. However, the precise content of the training interventions in these trials has varied. Much of this training has focused on increasing line managers’ awareness of mental health and improving their skills in how to support employees who are already experiencing poor mental health i.e., a secondary level intervention. There are few studies taking a primary prevention approach. Stansfeld and colleagues [24] focused on the line managers’ competencies that are required to prevent work-related stress [25]; however, their intervention did not improve employee outcomes, and line manager outcomes were not explored. The authors identified a need for greater focus on reflective and experiential learning to encourage behavior changes [24]. We have therefore endeavored to address this in our study through the rigorous development of a primary prevention intervention that includes reflection and experiential learning, delivered online.

Online approaches to workplace training are advocated to allow for greater flexibility in learning and increase workplace training capability [26]. E-learning platforms are a webspace or portal for educational content and resources that bring together all the resources a user needs into a single place. The use of e-learning has increased in recent years, and there is a plethora of platforms available (e.g., Coursera, Skillshare, Udemy, Codecademy, Edx, Pluralsight, Future Learn, Xerte, Moodle, and many more). The use of e-learning is escalating, following the rapid shift to online learning during the coronavirus (COVID-19) pandemic, potentially providing employers with a contemporary solution to increasing flexibility in workplace training and learning [27]. The increasingly competitive environment of modern times has led organizations to focus on how they can establish a sustainable competitive advantage [28]. Training delivered using e-learning offers a distinct, competitive advantage to employers. Since facilitators are not required, it is cheap to deliver and is seen to be cost effective [29]. Since travel is not required, this reduces personal risks for employees (e.g., lone travel, fear of failure in front of others) and the costs associated with course attendance (e.g., financial, and person-time), and reduces the environmental impact (e.g., from travel, printing, etc.). Online training is highly versatile and can be accessed at any time or place, therefore fitting with employees’ flexible work patterns and lifestyles. This approach allows ongoing access to resources, and therefore ongoing opportunities for learning and development. This flexibility of use allows users to have greater control over their learning, which can increase satisfaction and motivation for learners [28,30]. Training delivered through e-learning ensures a certain level of consistency and standardization in the materials being delivered. This cannot be guaranteed with face-to-face modes of delivery, which can be heterogenous in terms of content and delivery, as demonstrated elsewhere [20]. Digital training is also highly scalable, giving broad accessibility to employees in diverse geographical regions. For users, e-learning provides opportunities to link learning activities directly to the day-to-day experiences people have at work, and this opportunity to provide relevant learning activities (e.g., inclusion of reflection, experiential learning, or competency-based learning [30]) may motivate better engagement and retention of learners than other modes of training delivery [29].

Our overall aim, therefore, is to develop and test a digital intervention aimed at supporting employers in the prevention of mental ill-health at work. This primary prevention research was conducted as part of the Mental Health and Productivity Pilot (MHPP) [31], a wider initiative aimed at supporting employers across the Midlands region of the United Kingdom (UK) to improve the future of workplace mental health and wellbeing. The MHPP aims to (a) reduce the impact of poor mental health in the workplace and to remove barriers to employability and productivity, (b) help to reduce stigma around the workplace, (c) deliver evidence-based interventions. The Managing Minds at Work project is one of several interventions developed within the MHPP program, in which a digital interactive training program to support line managers will be developed and tested. The research will be conducted over several phases, including intervention development, feasibility testing, evaluation, and exploration of implementation processes. In this paper, we report the first stage of the research, which is the intervention development. The intervention is a digital training program called Managing Minds at Work (MMW). The aim of the intervention is to support line managers by increasing their knowledge, skills, and confidence for promoting and protecting the mental wellbeing of the people they manage.

## 2. Methods

### 2.1. Design

This study reports on a rigorous intervention development process adopting a collaborative participatory design [32], involving co-design (with users as partners for content development) and incorporating principles of user-centred design (conducting usability testing while considering human factors and ergonomics). Therefore, we adopted a person-centred approach as described in guidelines for the development of digital health interventions [33,34,35]. The study involved an ‘agile design’ process, which is a dynamic and flexible approach to co-design, adopting ‘kanban’ methods (where team members are allocated specific tasks without a pre-determined timeframe) as used in previous studies [36,37,38]. The approach draws on agile software development and generates directionally useful feedback that enables rapid learning and iterative improvements to both the content and technical design during the development of training materials. The intervention was then tested in a usability pilot study that was aligned to the Technology Acceptance Model (TAM [39]). This intervention development phase is an early stage of a registered trial (ClinicalTrials.gov Identifier: NCT05154019). The intervention development process follows the recommended principles and actions to allow links to be made in the future between intervention development processes and the subsequent success of the intervention [40]. Ethical approval for the research study was obtained from the institutional ethics review board (FMHS Ref: 299-0621).

### 2.2. Procedure

Intervention development took place between February and July 2021. The study adhered to the development phase of the Medical Research Council (MRC) framework for developing and evaluating complex interventions [41] and was mapped to the six core elements including: (i) context, (ii) developing and refining program theory, (iii) engaging stakeholders, (iv) identifying key uncertainties, (v) refining the intervention, and (vi) economic considerations (Table 1). Study reporting draws on relevant items from the ‘Checklist of questions to consider when planning, reporting, or analysing co-design activities’ [42] (Appendix A).

The intervention development was informed by the four critical elements of co-design outlined in Chisholm’s co-design model ([43] Figure 1). A virtual panel was established in May 2021 consisting of 10 expert stakeholders who contributed to the co-design of training content. As members of the virtual panel, stakeholders liaised with the project team through a range of methods including individual email correspondence, telephone discussion, and/or group discussion. Stakeholders included individuals from academic institutions, local authorities, and the mental health charity Mind and Public Health England. The stakeholder group had expertise in education and training, involving specialists in human resources, employment, and/or mental health, and people with lived experiences of mental ill-health and line management, including the supervision or management of employees with mental health concerns. The research team had expertise in occupational and health psychology, mental health, workforce and employment issues, digital health education, and equality, diversity, and inclusion. Collaborative working between the research team and stakeholders allowed for evidence-based (top-down) and experience-based (bottom-up) inputs to design. At each stage of the study, to merge inputs, we adopted existing bidirectional strategies to enable full co-design; such strategies have been used previously for digital intervention development, for example, in the creation of eHealth initiatives [44]. These include selecting (satisfying one need but not the other), combining (keeping multiple options in the design), integrating (designing a new and coherent functionality that serves both needs), and reframing (redefining perspectives in a way that dissolves the conflict).

After preliminary discussion with stakeholders, the initial outline of module headings was drafted by the research team as subject matter experts. Module headings were created using a ‘Wall Storm’ approach, in which key topic areas were proposed by team members and processed as a group through interactive virtual meetings. A storyboard was then co-created together with stakeholders. A storyboard is a document used to describe the text, visuals, audio, interactive elements, and navigation that will be used in a digital training program. The Managing Minds at Work storyboard was co-ordinated by the project researchers and co-created through multiple methods including a design charette (larger group meeting to sketch storyboard ideas), design jams (smaller group meetings to develop multiple iterations of user experiences e.g., as used in [43]), an online form for the collection of feedback data, and email communication. In this way, end-users and stakeholders worked together in playing an active role in design decision making which is ongoing [45,46] and highly flexible to maximise stakeholder engagement. The stakeholder group was a ‘virtual community of practice’ (vCoP; [47]); defined for the purpose of this study as ‘a group of people with a common interest who interact and share knowledge, using online platforms’. Stakeholders were invited to consider six broad areas:Order of modules and materials.Frequency and timing of module delivery.Content revisions (additions/removals).Targeting of information (e.g., specific/generic).What are the barriers that could prevent line managers from accessing/engaging in the training?What are the facilitators that could support line managers in completing the training?

This process was agile, with multiple co-creation and review activities being undertaken concurrently. Reviews and revisions were made iteratively, until a final storyboard was agreed that aligned with the program’s theory of change (see Section 3.1). The process included the consideration of future implementation of the training, and the relationship between interactive activities (e.g., reflections, quiz), mechanisms of change (e.g., improved understanding, acquiring manager competencies, learning communication skills), potential outcomes (e.g., improvement in communication or willingness to promote mental health), and context, including barriers and facilitators to use (e.g., availability of devices, ease of use).

Technical development was then undertaken by the project team, by creation of a prototype using Xerte, a multimedia authoring tool. In line with prototype development processes described elsewhere [48], design specifications were refined according to the level of importance assigned to them by stakeholders, and their views towards the likely acceptability, ease of use, technical difficulty, time required, and development costs. Following stakeholder review, revisions were made in five broad categories (Table 2).

An explanation of the value of the training was included in the introductory text to line managers to encourage institutional ‘buy-in’. Further revisions by module are shown in Appendix A. Stakeholders agreed to the final module topics (Figure 2; May 2021) and the final training program content (June 2021). Managing Minds at Work was then hosted on a Web platform in September 2021 and is ready for feasibility testing, a future phase of the research in which the training program will be delivered to managers in real-world settings, with an evaluation undertaken on ‘what’ was delivered, ‘how’ it was delivered, and how recipients respond.

Since the study was undertaken during the global coronavirus pandemic (COVID-19), we were guided by published practical actions to promote and maintain meaningful exchange with stakeholders in times of social distancing and lockdowns [49]. As described, we were able to sustain stakeholder exchange by offering alternative approaches to involvement and deducting which formats of engagement were suitable for different groups (Action 1: ‘know your stakeholders’). These options included brief, one-to-one conversations (Action 2: ‘strengthen stakeholder relationships’), emails, telephone contact (Action 3: do not go 100% digital’), and small-group meetings (Action 3: ‘re-think your offline methods’). To accommodate stakeholders, we allowed a significant period for development and included extended email correspondence to maintain the conversation (Action 4: ‘stay flexible and keep it simple’). Given that stakeholder input was gathered remotely, there is a potential for bias in that those involved with virtual stakeholder input may have been more likely to be familiar with technology and digital approaches to training (Action 5: ‘learn lessons for post-pandemic engagement’). Finally, given the pandemic-related increase in mental ill-health [50] and the impact of the pandemic on the labor market [51], we aim to consider the impact of the circumstances of the pandemic on intervention development and future feasibility testing (Action 6: ‘account for the COVID-19 circumstances in your research results’).

### 2.3. Usability and Pilot Testing

The intervention was pilot tested by the project team between October 2021 and March 2022, with the aim of testing the acceptability and usability of MMW to line managers in ‘real-world’ settings. The design was a single group intervention study with post-intervention data collection, aligned with the Technology Acceptance Model (TAM [39]) (Figure 3).

TAM is an information systems theory that models how users come to accept and use a technology. Participants were organizations (site participants) and line managers (individual participants). Since the entire study was conducted during a pandemic, which was a challenging time for organizations, we adopted opportunity sampling. There is no established consensus for the sample size required for usability testing, with a range of suggested options (e.g., 3, 4, 5, 7 ± 2, 16 ± 4) depending on the approach taken. We therefore opted to recruit the largest suggested sample size (16 ± 4) given the diversity of workplace training needs across organizations and sectors, and the changing needs in the context of the COVID-19 pandemic and its aftermath—it has been proposed that group size should typically be increased along with the study’s complexity and the criticality of its context [52].

## 3. Results

### 3.1. Managing Minds at Work Intervention

Intervention reporting aligns with the Template for Intervention Description and Replication (TIDieR) checklist and guide [53] (Appendix A). The resulting intervention is a workplace line manager training program called ‘Managing Minds at Work’. The intended audience is line managers who have direct managerial or supervisory responsibilities. The purpose of Managing Minds at Work is to develop line managers’ knowledge and confidence in preventing work-related stress and promoting mental health at work. The training is based on a theory of change, which is commonly used in the development of digital interventions [54]. The theory of change is a purposeful model of how the MMW intervention will contribute through a chain of early and intermediate outcomes to the intended result:

“Managing Minds at Work will develop line manager’s knowledge and confidence in preventing work-related stress and promoting mental health at work. This will be achieved through learning activities to increasing line managers’’ awareness of mental health (including legal requirements and employer responsibilities around work-related stress), encouraging the creation of psychologically safe working environments and work designs that promote mental wellbeing, and increasing managers’ competencies in preventing work-related stress and having open conversations about mental health in the workplace. Ultimately, the longer-term outcomes of this will be to reduce the prevalence of mental ill-health in working-age adults, and related economic burden of presenteeism and sickness absence to individuals, employers, and society”.

The program was developed using Xerte, a multimedia authoring tool focused on accessibility and simplicity. It consists of an introduction followed by five modules of online learning (Figure 2), accessed by URL using an individual user login and password. It is accessible via a secure website using a personal computer (PC), laptop, tablet, or mobile phone, although larger screen devices (e.g., PC or standard laptop) are recommended to maximize readability. It is intended to be accessed as a standalone training program, requiring no additional materials. The training materials are underpinned by an interactive and multi-media format. Each module includes evidence-based guidance, interactive tasks with opportunities for reflection, short videos, embedded links, and downloadable checklists. Each module ends with a ‘summary and next steps’ section, that provides a broad recap on key messages and key resources. A ‘next steps’ checklist at the end of each module invites the user to reflect on practical steps: (a) things you can do as a manager, (b) things to ask your organization to do, (c) things to ask your team to do.

The content is designed to be relevant to any line manager, in any organization and any sector. Therefore, it intentionally provides generic advice rather than sector-specific or tailored advice, which will allow for future intervention scalability. The modules are independent and stand-alone, although it is recommended that they are completed in succession (Introduction, then Module 1 through to Module 5), at a rate of 1 module per week, undertaken over a period of 6 weeks. Each module takes approximately 30 min to complete. The training is designed in this way so that learning can be undertaken flexibly during working hours. The training is self-led (requiring no prior knowledge or training to access and engage with it), so each participant can progress individually through each module at their own pace and at a time that suits them. Module content can be revisited at any time.

The final Managing Minds at Work content is shown in Figure 4. The management competencies that prevent work-related stress [55,56,57,58] are included in Module 3 (Figure 5). Approaches to having conversations with people about mental health at work are included in Module 5 (strategies shown in Figure 6). An example interactive activity is provided (Figure 7).

### 3.2. Usability and Pilot Testing Results

In October 2021, we approached three organizations that were taking part in mental health initiatives organised by MHPP (a regional consortium [31]) and therefore had ‘gatekeepers’ that were known to the project team. All organizations agreed to be pilot test sites and to approach managers employed within their organizations to invite them to take part in MMW piloting and usability testing. One of the organizations was from the third sector (a charity), one was from the private sector (a cooperative financial institution), and the other was from the public sector (an educational institution). One organization was small (0–49 employees), one was medium-sized (50–249 employees) and one was large (>250 employees). These gatekeepers provided information to all their employees about the usability study and invited them to directly contact the project team if they were interested in being involved. We aimed to recruit a sample size of 16 ± 4, and this was achieved. Twenty-seven managers expressed an interest in participating in the pilot study. Of these, 18 gave informed consent (67%), and were then invited to engage with the prototype training package and provide user feedback aligned with the Technology Acceptance Model (TAM; [39]) which is one of the most influential models of technology acceptance. Quantitative data were collected using a brief survey designed to assess the two primary factors influencing an individual’s intention to use new technology: perceived ease of use and perceived usefulness. This contained 11 question items, including 2 categorical items (yes/no), 5 Likert scale items (strongly agree, agree, disagree, strongly disagree), including 2 items that were reverse coded (i.e., phrased negatively), and 4 open-ended question items which allowed for free text responses to expand on attitudes towards the training, mental health promotion, and behavioral intentions (i.e., individual intention to use a particular technology that directly affects actual usage). Managers could choose whether they completed one or more modules as part of the pilot test.

Of the 18 managers, 12 (67%) completed at least 1 module, and a total of 37 survey responses were gathered across 5 modules (62% of a possible 60). This provided feedback from between 5–11 managers per module, which was considered adequate to establish their usability. Due to the pandemic context and impact on businesses during this time, we did not advocate that all the managers complete all five modules unless they wished to. Feedback related to module 1 (*n* = 11), module 2 (*n* = 10), module 3 (*n* = 5), module 4 (*n* = 6), and module 5 (*n* = 5). Age of managers ranged from 24–59 years (61% female). Participants had worked in a managerial role between 6 months and 30 years (two managers had supervised other people for 1 year or less, 11 managers had supervised other people for ≤10 years). Managers currently supervised up to 25 employees; half of the managers supervised ≤10 people (*n* = 9). Quantitative survey data were analyzed using descriptive statistics, in IBM SPSS Statistics for Windows, version 26.0 [59]. Findings are shown in Table 3.

Quantitative data and qualitative free text responses were then coded into the five TAM categories and narratively reported.

*External variable:* The MMW training was delivered as planned. However, while almost two-thirds of the managers could easily complete the materials within the working day, finding time to complete the modules was a barrier for over one-third of managers (37.8%).

*Perceived usefulness:* All the managers (100%) perceived that the module content was relevant to their managerial role. Most of the managers found the case examples relevant to their role, and only two offered reasons for non-relevance. One manager indicated that the case examples were not in their direct field of work, although they did acknowledge the similarities. Another raised an issue with a case example “putting the onus on the manager to control workload” when in some roles (e.g., in large organizations or expert teams), this is not possible at an individual level, and it may be more relevant to make workload decisions collaboratively as a team as to what can or cannot be delivered by the team. In such instances, acknowledging the views of employees ensures they have some level of input and control. It was proposed that future versions could contain more examples as resources for users to draw on. Almost two thirds of respondents indicated that they had learned something new from the training. Those that had not learned new information still found the materials relevant as they highlighted that the program served as a useful prompt or reminder to apply their knowledge in the workplace.

*Ease of use:* All the managers (100%) perceived that the modules were an appropriate length, and all but one of them found the content easy to understand and the online package easy to navigate (albeit one reported a technical issue). One manager proposed that audio recordings of the content would be a useful addition in the future.

*Attitude:* Managers referred to becoming more aware of the heterogenous impact of stressors on individuals after engaging with the training:

“It’s given me a better understanding of the different sources of stress for colleagues and subordinates. I’ll be sure to be more considerate of the different factors in my approach going forward”.

Some indicated that they had learned the importance of prioritizing their own mental wellbeing and role modelling positive behavior in the workplace, particularly by enacting self-care behavior, such as taking work breaks:

“I have underestimated how important it is that I look after myself. The information in the module makes sense in that you cannot support others if you aren’t in a good place yourself”.

“a good reminder of the need to take care of yourself”.

*Behavioral Intention:* Most of the managers indicated that they would be making practical changes following their use of the training program. Proposed actions were diverse but included monitoring stress levels within their organization (e.g., through use of a workforce stress survey). Several managers suggested that they would further review the sources of stress in their organization and seek to remove stressors and/or identify ways in which stress could be managed. Managers reported that they would use their learning to better recognize the signs of stress in colleagues, and their new understanding of their own responsibilities as a manager would help them to decide on actions to take. One manager commented that they would be more likely to act in a timelier way, and others indicated they intended to use the tools advocated within the materials (e.g., HSE Management Standards Indicator Tool [58]). Managers reported that they would re-consider job design and job demands in the future—in particular, providing staff with more clarity around their job roles, and focusing on whether job demands are both manageable and challenging.

“It was useful to learn about breaking down the job demands into areas I can easily understand. This will help me to help my team”.

There was a recognition that the COVID-19 pandemic had exerted a negative impacted on organizations and commonly resulted in higher workloads for staff. Managers committed to exploring the impact of workloads on staff stress levels and ways in which this could be mitigated, for example, the employment of additional staff, or reviewing workplace policies and behavior and actively encouraging behavior that supports wellbeing (e.g., advocating work breaks and self-care activities). One manager indicated they would be involving their employees more in problem solving rather than attempting to generate ‘top-down’ solutions themselves:

“I think, as the manager I need to have all the answers but it’s important to share problems with the team”.

Many of the proposed actions related to improving communication approaches between managers and other employees. For example, managers indicated that they would more regularly and consistently ‘check-in’ with staff, initiating more one-to-one conversations about mental health with colleagues, and consider the language they used to do this:

“I will make an effort to ensure that both my team and I are more conscious about the things we say and how we say them”.

“Made me rethink how to approach people - and the boundaries of things I might ask about”

Managers recognized the value of establishing the right workplace culture and environment to facilitate this. Several managers pledged actions related to improving the workplace culture, such as: “be kinder to colleagues”, “note down positive occurrences in the working day”, and “establish greater psychological safety within the workplace”:

“I had heard of psychological safety but didn’t really know what it was. I can now make more effort to ensure we have psychological safety in the team”.

“the more conversations I have, the easier it is to talk about mental health”.

Although the managers primarily referred to behavioral intentions in terms of the positive actions they would take in the future, one manager highlighted the potential challenges of implementing change, in terms of balancing the actions recommended within the training program with the expectations and demands of the business.

Overall, managers perceived the package to be easy to use, with high content readability (designed for a reading age of 12 years), high usability, and high fidelity (relevance to employment context). Some minor technical issues were raised when accessing the package from a device with a small screen (e.g., phone or tablet), resulting in a recommendation for the training to be accessed on a PC or laptop with a larger screen size. Strategies to maintain or enhance fidelity will be explored in a future feasibility trial.

## 4. Discussion

In this paper, we report on the theoretically informed development processes and pilot testing of a digital training program for line managers called Managing Minds at Work (MMW). The aim of MMW is to support employers in preventing mental ill-health and promoting good mental wellbeing at work. The key areas covered within the training are self-care, designing and managing work to promote mental wellbeing, management competencies to prevent work-related stress, developing a psychologically safe work environment, and having conversations about mental health at work. To our knowledge MMW is the first digital training to be co-created with line managers, using case examples and reflexive learning, which aims to equip line managers with the skills, knowledge, and competencies needed to design and operationalize a healthy workplace that promotes mental health.

Our findings indicate that managers perceive MMW to be appropriate and relevant for their needs, with high fidelity, ease of use, and relevance for managers from the public, private, and third sector. This is important given that managers play a key role in primary prevention, but many organizations offer no training for their staff [19], and there are few existing studies of interventions with a primary prevention focus [24]. Nevertheless, national and international guidance advocates the importance of a preventative approach and the essential role of line managers in mental health at work [60,61,62,63,64].

We included an element of reflexive and experiential learning within MMW training content as recommended in previous research [24] and opted for online training delivery given the high value placed on online, flexible learning approaches [26,27,28,29,30]—both elements were valued by stakeholders and managers for the ease of access to training and their assistance in supporting learning.

Co-creation of content using collaborative participatory design (e.g., [32,42]) has resulted in a training program that is directly relevant to the learning and support needs of line managers. The adoption of a user-centered active learning pedagogy with professional reflection on scenarios means that line managers are actively engaged in the learning process, gathering information, thinking, and problem solving, which is known to have better outcomes than more passive learning approaches [65]. Use of agile methodology has involved a rigorous development process [32,36] that allowed flexibility in virtual engagement with stakeholders and timings for the review processes. This was essential given that the study took place during the COVID-19 pandemic, a time of significant employment impact [66], when stakeholders were harder to reach, had less time, had priorities elsewhere, and/or government social distancing measures designed to stop the spread of the virus restricted face-to-face participatory activities [49].

The participatory design and rigorous co-creation processes have been described in detail, including the practical actions taken for managing the effects of the COVID-19 pandemic on participatory research. The importance of documenting intervention development processes, as a key research activity, has been outlined by O’Cathain and colleagues [40]. This is deemed to be an essential research activity which allows for judgements to be made about the quality of the process and outcome of future interventions and supports future replicability. This study is the first step in a research process which will test the feasibility and acceptability of MMW to managers in diverse employment settings, and then assess the effectiveness of MMW in improving outcomes for managers, employees, and organizations.

### Study Strengths and Limitations

A strength of the study was the rigorous process for content and technical development, involving participatory design with multi-method approaches to co-creation, adapted in line with guidance for conducting participatory research during the COVID-19 pandemic. Our intervention was based on a ‘theory of change’ and user-centered active learning pedagogy, and the research was theoretically informed, using Chisholm’s co-design model [43], and TAM [39], an information systems theory. The resource has been pilot tested across sectors (public, private, and third) and organization types and sizes, which has allowed us to explore the usability and fidelity of the training program prior to full feasibility testing. We did not collect demographic data for the individuals who participated in the expert panel to maintain confidentiality. In our empirical pilot study, the sample size was appropriate for usability testing, although not all managers were able to complete all the modules, due to the pressures and additional work demands related to the COVID-19 pandemic. Nevertheless, between 5–11 managers provided feedback on each module, and this was deemed sufficient to meet the aims of the pilot testing and usability study. Managers involved in our expert panel and usability study were primarily from office-based roles, and the training program is yet to be feasibility tested in other types of employment setting.

## 5. Conclusions

Managing Minds at Work is a new digital training program for line managers which aims to support employers in preventing mental ill-health and promoting good mental wellbeing at work. The intervention was deemed to have high usability and fidelity by managers from the public, private, and third sectors. The next step is to fully explore the feasibility and acceptability of the Managing Minds at Work intervention with managers in diverse employment settings, explore the mechanisms for its most effective delivery, and examine whether and how the intervention can be tailored for different types of organizations and integrated alongside existing polices, practices, and interventions.

## Figures and Tables

**Figure 1 ijerph-19-08006-f001:**
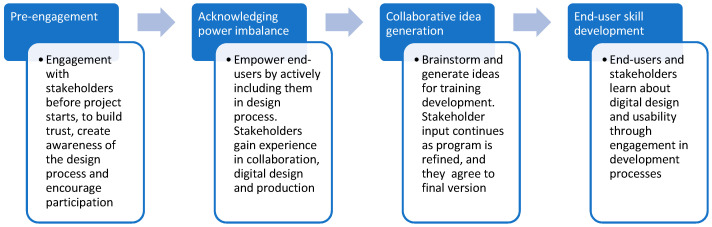
Application of Chisholm’s co-design model to Managing Minds at Work development.

**Figure 2 ijerph-19-08006-f002:**
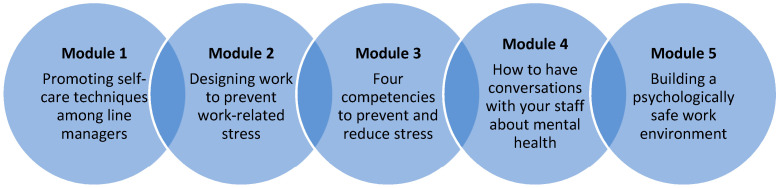
Overview of Managing Minds at Work modules.

**Figure 3 ijerph-19-08006-f003:**
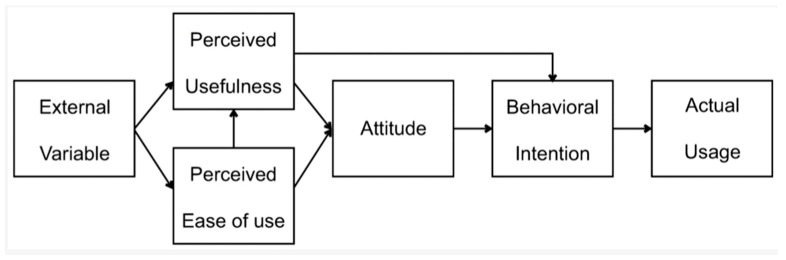
Technology Acceptance Model (from [39]).

**Figure 4 ijerph-19-08006-f004:**
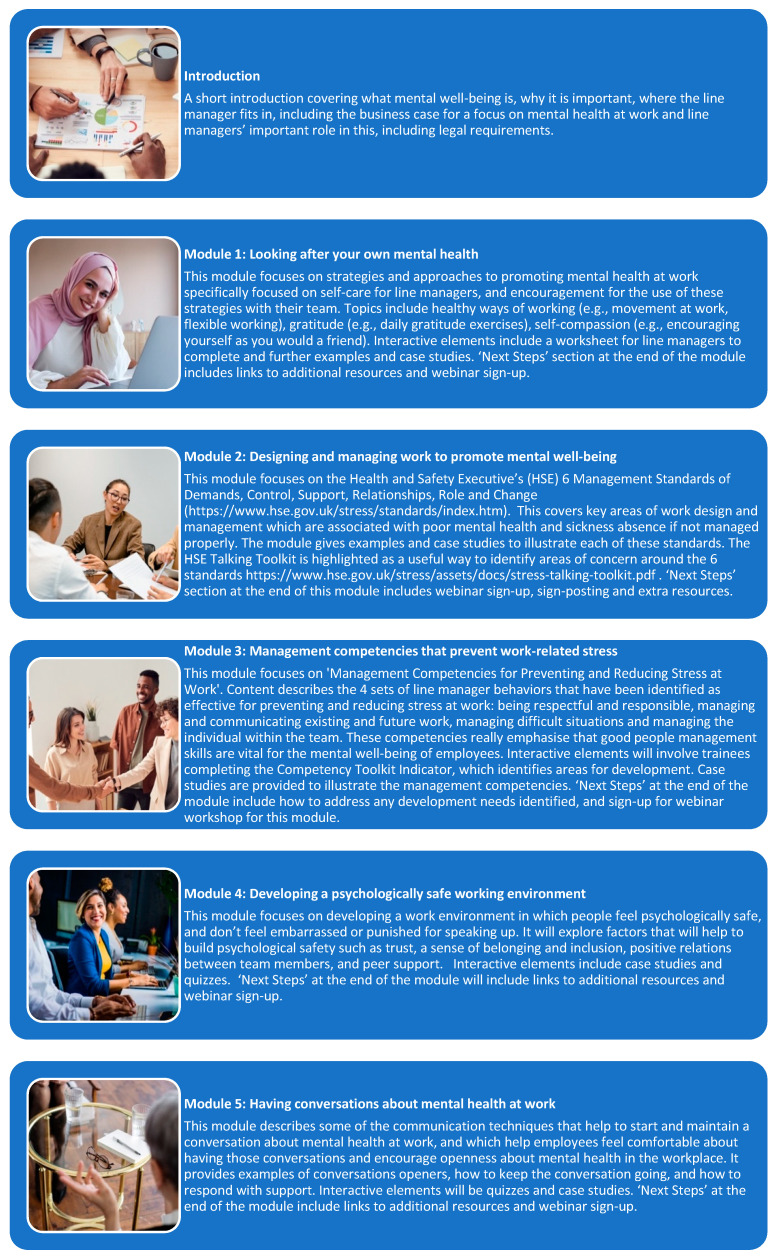
Managing Minds at Work Training Materials.

**Figure 5 ijerph-19-08006-f005:**
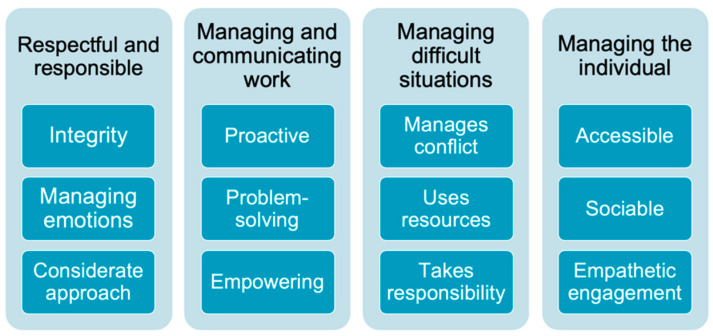
Management competencies that prevent work-related stress (module 3).

**Figure 6 ijerph-19-08006-f006:**
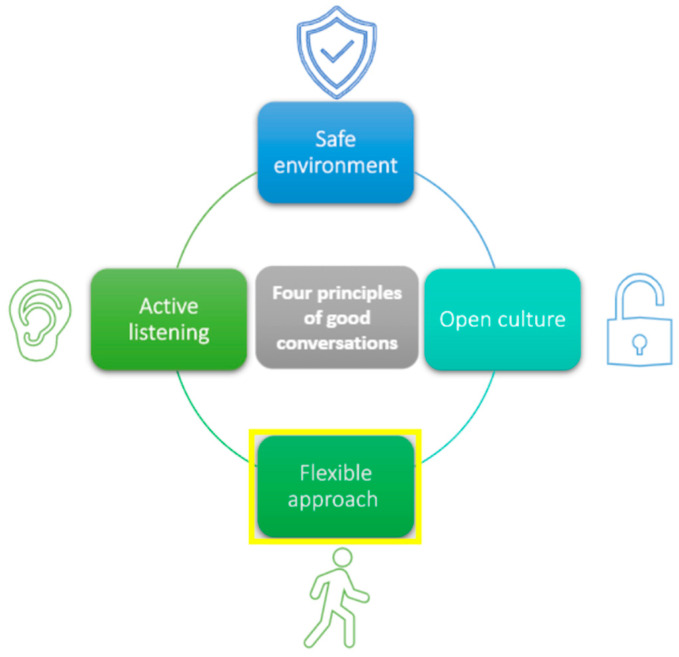
Strategies for having conversations with people about mental health at work (module 5).

**Figure 7 ijerph-19-08006-f007:**
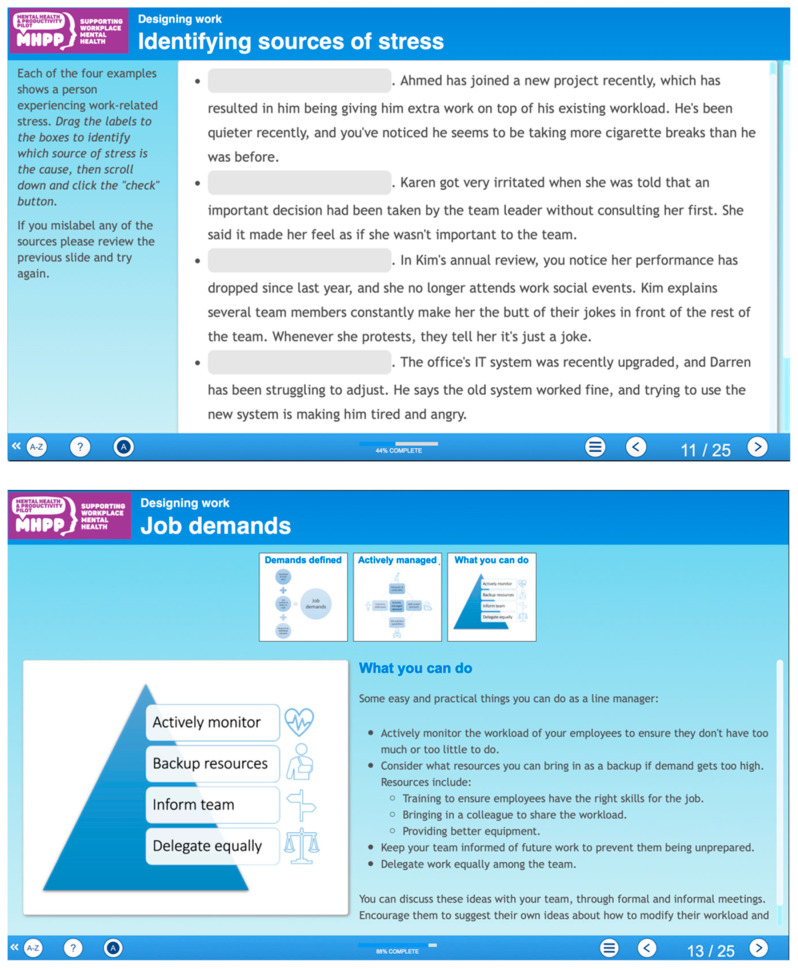
Example interactive activities (module 2) ^†^**.**
^†^ Two slides from the same module. First example requires users to identify which sources of stress are the cause of each of the examples by dragging four sources of stress shown along the bottom (job control; job demands; relationships at work; work and organizational change) to their appropriate grey boxes. Second example requires users to click on the image to gain practical examples.

**Table 1 ijerph-19-08006-t001:** Mapping of core elements of the MRC framework ^†^ to ‘Managing Minds at Work’ development.

Core Elements	Intervention Development Considerations
(i) Context	Context of the employment settings and job role of the line manager may influence the way in which the intervention is accessed and used. Content must be relevant across employment settings (sector, size, and type of organization) for line managers at all levels of the hierarchy.
(ii) Developing and refining program theory	Program theory was established prior to intervention development with the involvement of diverse stakeholders and based on evidence and theory from relevant fields. This focused on identifying the key areas of line managers’ influence in preventing poor mental health, specific actions associated with these, and the likely outcomes. Can be refined during successive phases to inform transferability of the intervention across settings.
(iii) Engaging stakeholders	Collaborative participatory approach involved stakeholders at every stage: development of program theory, co-creation of intervention content, iterative peer review, and revisions. Multiple methods for stakeholder engagement were triangulated and included individual/group discussion, online data collection, and email feedback to share and develop ideas. Participation must be both emergent and ongoing (long term).
(iv) Identifying key uncertainties	Uncertainties related to design and delivery: timescale for development of the intervention, appropriateness of the level of language complexity, most appropriate format for delivery. Potential challenges with engaging stakeholders in research intervention development during a global pandemic. Consideration of the global public health and economic impact of a global pandemic on intervention content and future implementation.
(v) Refining the intervention	Using an agile approach, stakeholder consultation and review is an iterative process, allowing for continuous delivery and a resource-efficient approach to toolkit development.
(vi) Economic considerations	Stakeholder input supported by charitable bodies and professional input via the wider MHPP program. Specific costs for intervention web-hosting and individual user logins.

^†^ Medical Research Council (MRC) framework for developing and testing complex interventions.

**Table 2 ijerph-19-08006-t002:** Revisions to Managing Minds at Work content and design during co-creation and review.

Category	Summary of Stakeholder Revisions
Presentation of materials	cover page, colors, logos, image diversity, balance of text, activities and images, use of bullet points, signposting, typographical errors
Functionality	font and illustration size, scrolling, web links, transcript availability, use of hyperlinks, video quality
Clarity of information	definition of terms, not assuming prior knowledge or skills, rephrasing, additional explanation, removal of repetition
Additional resources	adding confidential helplines, downloadable resources page
Incentives for completion	module-by-module completion for flexibility, provision of feedback or explanation for incorrect answers on tasks, encouragement to revisit tasks, observable progression points, confirmation of completion, reminders and encouragement for completion, downloadable certificate
Consideration of current context	relevant to virtual and remote working due to the global coronavirus pandemic

**Table 3 ijerph-19-08006-t003:** Usability testing and associated TAM constructs.

Question Item (N = 37 Responses ^+^)	TAM Construct	*n* (%)
Knowledge attainment		% Yes
Did you learn anything that you did not know before?	U	24 (64.9)
Content Relevance		% Yes
Did you think the module content was relevant to your managerial role?	U	37 (100)
Case example relevance		*n* (%) strongly disagree, or disagree
The examples provided throughout the module were not relevant to my role as a manager	U	30 (81.1)
Ease of understanding		*n* (%) strongly disagree, or disagree
I found some of the information presented in the module difficult to understand	E	36 (97.3)
Usability		*n* (%) strongly agree, or agree
The online module was an appropriate length	E	37 (100)
The online module was easy to navigate	E	36 (97.3)
Barriers to use		*n* (%) strongly agree, or agree
It was easy to find the time to complete this module	EV	23 (62.2)

TAM Technology Acceptance Model; U, perceived usefulness; E, perceived ease of use; EV, external variable. ^+^ 37 survey completions from 12 managers.

## Data Availability

This was an intervention development study that involved co-creation and usability testing with participants rather than the collection of research data. However, further details on the development activities, stakeholder feedback, and usability testing are available from the authors on reasonable request.

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
