# Peer review of "Managing Minds at Work: Development of a Digital Line Manager Training Program"

_ijerph, 2022, doi:10.3390/ijerph19138006_

Round 1

Reviewer 1 Report

Dear Author(s),

Thank you for your paper. I would like to underline that I found the paper interesting and with several aspects and elements of novelty. 

These are very minor revisions suggested, as I believe this paper would be a great fit for the journal:

  • The reading, in general, is not very easy. The text does not flow, passing from one citation to another. Probably they could be better connected.
  • More careful use of citations. For example, lines 39-41 make a general statement, while the Deloitte report is very specific about the pandemic timeline.
  • Paper “covers five broad areas” (line 26) based on the non-scientific publication of Deloitte (49-54). Could be relevant to support them in double-peered reviewed progressive literature.
  • Can be better underlined the difference between “holistic approach” (line 54) and “feasibility testing” (line 21). Could be important to explain what the authors mean by “feasibility testing” and the reason why some factors are not considered. For example, the legal factor is not mentioned.
  • Type of “e-learning platforms” (line 84) with examples and differentiation characteristics. Including the idea of competitive advantage of Managing Minds at Work.
  • What is the role of the expert stakeholders (line 132)? How their contribution was collected? Was it a discussion in the group (line 147)?
  • What do authors understand as eHealth (line 141) and why it is important for this context?
  • What do you mean by “an online 156 data collection” (156-157)
  • Figure 2 (192) are two different figures with the source missing.
  • What were the criteria for organization and manager selection (line 196)? How they were divided?
  • The literature review is not very rich and systematic. Maybe the title itself should be “Managing Minds at Work Case Study”?

Thank you again for your work and I wish you all the best in your future research!

Author Response

I would like to underline that I found the paper interesting and with several aspects and elements of novelty. These are very minor revisions suggested, as I believe this paper would be a great fit for the journal.

Thank you for these positive comments.

The reading, in general, is not very easy. The text does not flow, passing from one citation to another. Probably they could be better connected.

We have checked and revised the flow of writing – this has also involved revising the results section, and re-writing both the introduction and discussion. We have endeavoured to link sections together better (e.g., in the new introduction text).

More careful use of citations. For example, lines 39-41 make a general statement, while the Deloitte report is very specific about the pandemic timeline.

We have re-written the introduction. This has involved adding new citations and improving the flow of writing. We have retained the reference to the Deloitte report but have added HSE (2019) report which appears first, and other citations.

Paper “covers five broad areas” (line 26) based on the non-scientific publication of Deloitte (49-54). Could be relevant to support them in double-peered reviewed progressive literature.

Thank you for this comment. We have re-written the introduction and although the Deliotte report has been retained, we have provided additional high-quality and recent citations. We believe this has strengthened the rationale.

Please note that the paper itself does not cover 5 broad areas. Rather, the Managing Minds and Work training programme that we developed (not the paper in which we report on the development processes) covers five broad areas: (i) promoting self-care techniques among line managers; (ii) designing work to prevent work-related stress; (iii) management competencies to prevent and reduce stress; (iv) having conversations with employees about mental health; (v) building a psychologically safe work environment.

Can be better underlined the difference between “holistic approach” (line 54) and “feasibility testing” (line 21).

These terms are referring to very different things.

‘Holistic approach’ relates to the underlying rationale for our focus – it is referring to the need for primary, secondary, and tertiary level interventions in the workplace setting. We have outlined what these three levels of intervention are, and the description and use of the term holistic is drawn from the joint report from the European Foundation for the Improvement of Living and Working Conditions and the European Agency for Safety and Health at Work (with we have cited). We have made some adjustments to the wording in this paragraph to be completely clear what this means and to strengthen signposting to the European report citation.

‘Feasibility testing’ relates to research methodology – it is one of the steps taking in evaluating complex interventions, according to the MRC guidelines that we are following. Feasibility testing will be the next step (ie., our next study) that comes AFTER intervention development (ie., our current study). Therefore, we have referred to it as an important next step but have not expanded on feasibility study methodology as we are not reporting on that here.

This is already stated clearly throughout the paper:

  • Our abstract aim: “The aim of the study was to describe the design and development of the Managing Minds at Work digital training programme, prior to feasibility testing”

  • Our abstract conclusion: “The next step is to explore the feasibility and acceptability of this intervention with line managers in diverse employment settings”.

  • Our introduction, rationale: “The research will be conducted over several phases, including intervention development, feasibility testing, evaluation, and exploration of implementation processes. In this paper, we report the first stage of the research, which is the intervention development”.

  • Our methods: “Managing Minds at Work was then hosted on a Web platform in September 2021 ready for feasibility testing, a future phase of the research”.

  • Our methods: “Strategies to maintain or enhance fidelity will be explored in a future feasibility trial”.

  • Our discussion: “…we aim to consider the impact of the circumstances of the pandemic for the intervention development and future feasibility testing”

  • Our conclusions: “The next step is to explore the feasibility and acceptability of the Managing Minds at Work intervention with line managers in diverse employment settings, explore mechanisms for most effective delivery, and examine whether and how the intervention can be tailored for different types of organisations, and integrated alongside existing polices, practices and interventions:”.

Could be important to explain what the authors mean by “feasibility testing” and the reason why some factors are not considered. For example, the legal factor is not mentioned.

We are not reporting on a feasibility study here. We are following MRC guidelines for the development of complex interventions. As such, an early stage of the research should be ‘intervention development’ (as we are reporting in the current study). Feasibility testing will be the next stage of the research (not included in the current study) and will follow this intervention development stage.

However, for any readers who are not familiar with the MRC guidelines, in addition to our original description of the guidelines in the methods, we have included further text to clarify the purpose of feasibility testing (which we will undertake next):

“…ready for feasibility testing, a future phase of the research in which the training pro-gramme will be delivered to managers in real-world settings, and evaluation undertaken on ‘what’ was delivered, ‘how’ it was delivered, and how recipients respond”.

Thank you for noticing the absence of reference to legal factors. The legal requirement for organisations is already included within the intervention content aimed at managers, but we have now referred to this in the intervention description and figure 3 (introduction section of the intervention).

Type of “e-learning platforms” (line 84) with examples and differentiation characteristics. Including the idea of competitive advantage of Managing Minds at Work.

Thank you for this important and useful feedback. We have added a new section to the introduction (and 3 additional citations) and this now better sets the context:

“Online approaches to workplace training are advocated to allow for greater flexibility in learning and increase workplace training capability [27]. E-leaning platforms are a webspace or portal for educational content and resources that brings together all the resources a user needs into a single place. The use of e-learning has increased in recent years and there is a plethora of platforms available (e.g., Coursera, Skillshare, Udemy, Codecademy, Edx, Pluralsight, Future Learn, Xerte, Moodle, and many more). The use of e-learning is escalating, following the rapid shift to online learning during the coronavirus (COVID-19) pandemic, potentially providing employers a contemporary solution to increasing flexibility in workplace training and learning [28]. The increasingly competitive environment of modern times has led organizations to focus on how they can establish sustainable competitive advantage [29]. Training delivered using e-learning offers a distinct, competitive advantage to employers. Since facilitators are not required, it is cheap to deliver and is seen to be cost effective [30]. Since travel is not required, this reduces personal risks for employees (e.g., lone travel, fear of failure in front of others) and costs associated with course attendance (e.g., financial, and person-time), and reduces environmental impact (e.g., from travel, printing, etc.). Online training is highly versatile and can be accessed at any time or place, therefore fitting with employees’ flexible work patterns and lifestyles. This approach allows ongoing access to resources, and therefore ongoing opportunities for learning and development. This flexibility of use allows users to have greater control over their learning, which can increase satisfaction and motivation for learners [29,31]. Training delivered through e-learning ensures a certain level of consistency and standardization in the materials being delivered. This cannot be guaranteed with face-to-face modes of delivery which can be heterogenous in terms of content and delivery, as demonstrated elsewhere [21]. Digital training is also highly scalable, giving broad accessibility to employees in diverse geographical regions. For users, e-learning provides opportunities to link learning activities directly to the day-to-day experiences people have at work, and this opportunity to provide relevant learning activities (e.g., inclusion of reflection, experiential learning, or competency-based learning [31]) may motivate better engagement and retention of learners that other modes of training delivery [30].”

What is the role of the expert stakeholders (line 132)? How their contribution was collected? Was it a discussion in the group (line 147)?

They contributed to co-creating the content through a range of methods (which we kept flexible due to the challenges of engaging stakeholders in research during the pandemic which is discussed elsewhere in the paper). We have added some text to clarify this at the outset. The remainder of the section gives all the details.

“A virtual panel was established in May 2021 consisting of 10 expert stakeholders who contributed to the co-design of training content. As members of the virtual panel, stakeholders liaised with the project team, through a range of methods including indi-vidual email correspondence and/or telephone discussion, and/or group discussion.”

What do authors understand as eHealth (line 141) and why it is important for this context?

eHealth is the use of information management and information and communication technology (ICT) to assist with finding, using, recording, managing, and transmitting information to support health care, in particular, to make decisions about patient care.

We are not referring to our own intervention as ehealth here, instead we are giving an example of bidirectional co-creation strategies that have been used previously in the development of digital interventions, and we used an e-health example for this.

We have adjusted the wording in this paragraph to clarify our meaning:

“At each stage of the study, to merge inputs, we adopted existing bidirectional strategies to enable full co-design; such strategies have been used previously for digital intervention development, for example, in the creation of eHealth initiatives [33]. These include…”

What do you mean by “an online data collection” (156-157)

This phrase had the word ‘form’ at the end (“an online data collection form”).

To reduce misunderstanding, we have changed the phrase to “an online form for the collection of feedback data”.

Figure 2 (192) are two different figures with the source missing.

To avoid confusion, we have removed the second part of Figure 2 as it included some replicated information. Figure 2 now only shows 5 blue circles, indicating the key module topic areas. There is no source information because the authors created this specifically for this manuscript.

What were the criteria for organization and manager selection (line 196)? How they were divided?

We have included a new section in the methods on our sampling and recruitment (as below), and then provided the results for the usability and pilot study separately.

“Usability and pilot testing

The intervention was pilot tested by the project team between October 2021 and March 2022, with the aim of testing acceptability and usability of MMW to line managers in ‘real-world’ settings. The design was a single group intervention study with post-intervention data collection, aligned with the Technology Acceptance Model (TAM [40]) (figure 3).

TAM is an information systems theory that models how users come to accept and use a technology. Participants were organizations (site participants) and line managers (in-dividual participants). Since the entire study was conducted during a pandemic, which was a challenging time for organizations, we adopted opportunity sampling. There is no established consensus for the sample size required for usability testing, with a range of suggested options (e.g., 3, 4, 5, 7±2, 16±4) depending on the approach taken. We therefore opted to recruit the largest suggested sample size (16±4), given the diversity of workplace training needs across organizations and sectors, and the changing needs in the context of the COVID-19 pandemic and its aftermath - it has been proposed that group size should typically be increased along with the study’s complexity and the criticality of its context [53].”

We have included demographic characteristics for the pilot sample.

The literature review is not very rich and systematic. Maybe the title itself should be “Managing Minds at Work Case Study”?

We have not changed the title as this is not a case study – we have not followed case study methodology. Instead, it is an intervention development in line with the MRC framework and this is clear in the methods.

However, to respond to this comment we have completely re-drafted the introduction and discussion, and also included empirical data in the results (usability and pilot testing).

Thank you again for your work and I wish you all the best in your future research!

We appreciate this comment, and many thanks for taking the time to review our work.

Reviewer 2 Report

Dear authors,

Thank you for the opportunity of reading your article. The topic is interesting and relevant for the working environment field and public health. The article’s aim is to propose an instrument for managing mental illness by line managers by developing using agile methodology to co-create processes with a high practical contribution to the management and mental illness field.

The main strengths of the paper consist in the topic of the research and the practical framework based on the elements of 24 the Medical Research Council, respectively the managerial contributions of the paper.

The weaknesses of this article are found in the research methodology, presentation of the results and conclusions. The methodology description is not clear. Some of the following points could be followed for the paper to be improved:

  1. Is the agile research methodology qualitative? or quantitative?
  2. How was the research conducted comparative to the previous research on mental health management?
  3. There is no specification of any statistical procedures specified in the selection of the respondents – stakeholders or at least a clear criteria.
  4. There are mentioned 10 expert stakeholders from different fields, could it be provided a table or a list of the specialization of each stakeholder separately? Are this stakeholders managers dealing with mental illness among the workers they supervise?
  5. The resulted modules are important for managerial usage, but it is not clear the contribution to an academic theory or to the academic research methodology of management or mental health fields.
  6. Theory of change is mentioned in the paper. Could the authors provide more insight on Theory of Change and how it provides a framework for the study in the introduction, research design and results?
  7. It is not clear how the strategies from page 9 were obtained, or the content in the slides from pages 9 and 10. What was the methodology behind the content in the slides and obtained strategies? What theory and results did the authors based on proposing these strategies?
  8. The results and conclusions are vague and do not put the research results in relation with previous academic research. Are there any similar programs previously conducted? Are there any similar researches on managing mental health at work? The managerial contributions are very clear. They could be improved by mentioning implications on COVID-19 pandemic, given the fact research was conducted in the pandemic context. The theoretical contributions of the paper are not very well pointed, and they are not mentioned in the conclusions section. Authors could provide some arguments on theoretical contributions of the paper to the academic research.
  9. References on previous academic research on mental health at work and the management of mental health are few, more studies on mental health management and COVID-19 pandemic could be added.

The article brings good contribution on mental illness management practice and management, theoretical and methodology contributions should though be better emphasized.

Author Response

Thank you for the opportunity of reading your article. The topic is interesting and relevant for the working environment field and public health.

Thank you for this positive comment.

The main strengths of the paper consist in the topic of the research and the practical framework based on the elements of 24 the Medical Research Council, respectively the managerial contributions of the paper.

Thank you for this positive comment.

The weaknesses of this article are found in the research methodology, presentation of the results and conclusions. The methodology description is not clear. Some of the following points could be followed for the paper to be improved:

1. Is the agile research methodology qualitative? or quantitative?

This may be a misinterpretation as we are reporting on an intervention design and development process, and agile is an approach to e-learning development (commonly used for software development). We used agile methods in developing our training content and its technical presentation in the form of an e-learning package. We have clarified the text at the point we first mention agile:

“The study involved an ‘agile design’ process, which is a dynamic and flexible approach to co-design, adopting ‘kanban’ methods (where team members are allocated specific tasks without a pre-determined timeframe) as used in previous studies [37-39]. The ap-proach draws on agile software development and generates directionally useful feedback that enables rapid learning and iterative improvement to both content and technical design, during the development of training materials. The intervention was then tested in a usability pilot study, aligned to the Technology Acceptance Model (TAM [40]).”

However, the pilot and usability testing of the intervention involved collection of data which was mixed-methods (survey with quantitative categorical items and free text qualitative responses) and this has now been clarified as more detail has been added about the pilot test - see methods and results sections).

2. How was the research conducted comparative to the previous research on mental health management?

We have completely re-written the introduction and revised the discussion to better set the context of the training development in the light of the literature.

3. There is no specification of any statistical procedures specified in the selection of the respondents – stakeholders or at least a clear criteria.

Statistics would not be appropriate in the reporting of the intervention development, as the stakeholders were engaged in co-creation processes to develop content for the training programme using agile methods, and this was achieved through systematic and rigorous stakeholder engagement processes (not quantitative research data collection).

However, we have added significantly more detail about our pilot study that now includes reporting of quantitative survey data (using descriptive statistics) to examine usability and relevance of the intervention, which has been reported alongside qualitative data.

 4. There are mentioned 10 expert stakeholders from different fields, could it be provided a table or a list of the specialization of each stakeholder separately? Are the stakeholders managers dealing with mental illness among the workers they supervise?

We have already provided the expertise of the group of 10, although to address this comment, we have been more explicit that this includes people who have managed staff with mental health concerns.

“Stakeholders included individuals from academic institutions, local authorities, the mental health charity Mind and Public Health England. The stakeholder group had expertise in education and training, specialists in human resources, employment and/or mental health), lived experience of mental ill-health and line management, including supervision or management of employees with mental health concerns. The research team had expertise in occupational and health psychology, mental health, workforce and employment issues, digital health education, and equality, diversity, and inclusion. Collaborative working between the research team and stakeholders allowed for evidence-based (top-down) and experience-based (bottom-up) input to design.”

We have reflected on the request for a table of list describing each individual but have actively chosen not to include this. First, it would lead to duplication of text since all the stakeholders have expertise across multiple areas. Second, we believe that it does not add anything to the reader’s understanding of the expertise of those inputting into the co-creation processes since it is the overall expertise of the group that is most relevant here. Third, it would increase the risk of identification of individuals participating in this process, whom we have not named for reasons of confidentiality.

However, we have now provided demographic characteristics for the managers that took part in our pilot and usability test (section 3.2).

5. The resulted modules are important for managerial usage, but it is not clear the contribution to an academic theory or to the academic research methodology of management or mental health fields.

We have referred to several theories at various points of the manuscript (main manuscript, and in supplementary files). Our intention is not to develop theory, but since this manuscript reports on the development and testing of MMW, we demonstrate how the development (and testing) processes were informed by theory. i.e., we primarily link our work to the academic research methodology of intervention development. Since the focus of this paper is on the development processes for a digital intervention, we deem this focus (e.g., participatory design, co-creation, pedagogy, technology acceptance) to be more appropriate than links to management and mental health at this stage, which will be made more explicit in the future feasibility and effectiveness studies.

Our approach to intervention design and development was informed by theory – Chisholm’s four critical elements of co-design. This theory is applied in the methods and figure 1 (and supplementary file 1).

The content for the intervention was informed by a Theory of Change as described in the ‘Managing Minds at Work Intervention” section (and supplementary file 1), and user-centred active learning pedagogy.

The pilot testing and usability study aligned with the Technology Acceptance Model (TAM). TAM is an information systems theory that models how users come to accept and use a technology.

We have ensured that reference to theory is clearer in the manuscript, supplementary files, and conclusions.

6. Theory of change is mentioned in the paper. Could the authors provide more insight on Theory of Change and how it provides a framework for the study in the introduction, research design and results?

The theory of change is a purposeful model of how MMW intervention will contribute through a chain of early and intermediate outcomes to the intended result. It is not a framework for the study (development and pilot testing) but is a theory of how we expect the intervention to work. We have made this clearer in the text and included a citation. It is also referred to in supplementary files.

“The training is based on a theory of change, which is commonly used in the development of digital interventions [55]. The theory of change is a purposeful model of how the MMW intervention will contribute through a chain of early and intermediate outcomes to the intended result.”

 7. It is not clear how the strategies from page 9 were obtained, or the content in the slides from pages 9 and 10. What was the methodology behind the content in the slides and obtained strategies? What theory and results did the authors based on proposing these strategies?

The study adhered to the development phase of the Medical Research Council (MRC) framework for developing and evaluating complex interventions and mapped to the six core elements including: (i) context, (ii) developing and refining programme theory, (iii) engaging stakeholders, (iv) identifying key uncertainties, (v) refining the intervention and (vi) economic considerations (Table 1). We applied Chisholm’s co-design model and engaged stakeholders through multiple co-creation and peer review processes to establish an intervention theory of change, and intervention content (methodology as detailed in the methods). Pilot and usability testing was based on the Technology Acceptance Model (from Davis, 1989) also detailed in the methods.

 8. The results and conclusions are vague and do not put the research results in relation with previous academic research. Are there any similar programs previously conducted? Are there any similar researches on managing mental health at work?

We have re-drafted both the introduction and the discussion to address this comment.

The managerial contributions are very clear. They could be improved by mentioning implications on COVID-19 pandemic, given the fact research was conducted in the pandemic context.

We have referred to the COVID-19 pandemic at various points in our paper, to reflect on the context in which we conducted the study, the impact on managers, and the impact of the pandemic on our decisions with regards the research approach and procedures.

The theoretical contributions of the paper are not very well pointed, and they are not mentioned in the conclusions section. Authors could provide some arguments on theoretical contributions of the paper to the academic research.

We have referred to theory at various points of the manuscript. Our intention is not to develop theory, but since this manuscript reports on the development and testing of MMW, we demonstrate how the development (and testing) processes were informed by theory.

Our approach to intervention design and development was informed by theory – Chisholm’s four critical elements of co-design. This theory is applied in the methods and figure 1.

The content for the intervention was informed by a Theory of Change as described in the ‘Managing Minds at Work Intervention” section, and user-centred active learning pedagogy.

The pilot testing and usability study aligned with the Technology Acceptance Model (TAM). TAM is an information systems theory that models how users come to accept and use a technology.

We have ensured that reference to theory is clearer in the manuscript and in the conclusions section.

9. References on previous academic research on mental health at work and the management of mental health are few, more studies on mental health management and COVID-19 pandemic could be added.

Thank you for this comment. This should now be addressed as we have re-written the introduction.

The article brings good contribution on mental illness management practice and management, theoretical and methodology contributions should though be better emphasized.

Thank you for the positive comment. We believe our methodology contributions and the alignment of our research approach with theory is thoroughly described, and we hope that the critical comment has been addressed now through our revisions.

Reviewer 3 Report

Dear Authors!

The endeavour of facilitating line managers' involvement into the mental health problems of workers is praiseworthy. The topic at large would belong to the field of public health if it presented results from a test or pilot project. However, with all its merits, the paper remains on the level of solely describing the content of a highly important digital intervention module designed for line managers. As such, the paper has a section on results which contains no empirical results. Implicitly it states that the result is the production of the digital intervention tool itself, which is not a result based on any kind of empirical evidence. The journal IJERPH is focused on publishing a range of topics and allows for a range of methods. This purely technical paper, however, does not fit into the scope of the journal, as it contains no empirically underscored results. A valid approach would be, for instance, to test the prevention skills or competences of those line managers who used the training with respect to the mental health of their employees, and to compare it with the competences of those managers who did not participate at the training. This would be an example for a controlled trial. Other empirical testing methods could also be imaginable, but without such a part, the paper cannot be accepted for publication, even though the idea is original and highly important. Please consider adding some empirical elements to the research and resubmitting to this or to another journal.

Best wishes and good luck,

reviewer

Author Response

The endeavour of facilitating line managers' involvement into the mental health problems of workers is praiseworthy.

Thank you for this positive comment.

The topic at large would belong to the field of public health if it presented results from a test or pilot project.

We conducted usability testing with 18 managers (of which 12 responded, providing 37 feedback surveys across the 5 modules). We have described the method and provided the results (both quantitative and qualitative) with significantly more information.

However, with all its merits, the paper remains on the level of solely describing the content of a highly important digital intervention module designed for line managers. As such, the paper has a section on results which contains no empirical results. Implicitly it states that the result is the production of the digital intervention tool itself, which is not a result based on any kind of empirical evidence.

This is a paper reporting on the development of an intervention, which is an important research stage outlined in the MRC guidelines for the development of complex interventions. Therefore, the main outcome of the paper should be the description of the developed intervention.

The ‘Framework of actions for intervention development’ outlined by O’Cathain et al (2019) emphasises the importance of publication of intervention development projects as a key research activity:

“Write up the intervention development process so that judgements can be made about the quality of the process, links can be made in the future between intervention development processes and the subsequent success of interventions, and others can learn how it can be done” (O’Cathain et al, 2019)

Citation: O'Cathain A, Croot L, Duncan E, et al. Guidance on how to develop complex interventions to improve health and healthcare. BMJ Open. 2019;9(8):e029954. Published 2019 Aug 15. doi:10.1136/bmjopen-2019-029954

However, we have also conducted pilot and usability testing which is empirical evidence and this is now much more clearly reported, with quantitative and qualitative empirical results.

The journal IJERPH is focused on publishing a range of topics and allows for a range of methods. This purely technical paper, however, does not fit into the scope of the journal, as it contains no empirically underscored results. A valid approach would be, for instance, to test the prevention skills or competences of those line managers who used the training with respect to the mental health of their employees, and to compare it with the competences of those managers who did not participate at the training. This would be an example for a controlled trial.

This is no longer a purely technical paper - we have now included empirical data from our usability and pilot testing which should address this concern.

We agree that the intervention should be tested in a controlled trial, but it is not appropriate to move directly to this type of study. Our intervention development study is an early stage in the MRC framework for the development of complex interventions. Once the current development study is complete, we will then move to feasibility testing. If feasibility testing shows the intervention to be feasible and acceptable, then we would then progress to a controlled trial. However, each stage of the research is an important stage that will be completed and published separately, and in succession. 

Other empirical testing methods could also be imaginable, but without such a part, the paper cannot be accepted for publication, even though the idea is original and highly important. Please consider adding some empirical elements to the research and resubmitting to this or to another journal.

We have included empirical data in the results section (approach described in the methods).

Reviewer 4 Report

Dear authors, thank you for the opportunity to get acquainted with your interesting and unique developments. Your article has a number of advantages:

The introduction substantiates in detail the relevance of the study and the scientific study of the issue.

The design of the study, the very procedure for developing the program is spelled out in great detail, both the principles of development, which allow assessing the scientific validity of the author's approach, and the content of the program, are covered.

The results are described in detail in terms of describing the content of the program, its structure and visual presentation of materials for managers.

At the same time, it is necessary to make adjustments to some sections of your article for better understanding:

1. The Usability testing section needs to be supplemented with more complete information about the sample of pilot testing: age, seniority of a managerial position, professional area, how many employees he manages, etc. It is also necessary to include information on what questions were asked to the participants of the pilot study regarding the convenience of the program, their assessment of the acquired knowledge and etc.

2. In the results section, it is necessary to add information in%, how many people from the pilot sample, which gave answers to questions about the convenience and usefulness of this program. And not only in a general way to present the main conclusions (as indicated in lines 203-209). Because until there is a reliable understanding of the effectiveness of the use of the program during the pilot phase and it is not indicated what specific changes were made to the program after the results of the pilot project. We need both statistics and qualitative conclusions. So far, the conclusions and discussion of the results are more of a theoretical nature.

3. It is necessary to single out in a separate section the limitations of the study and further directions for the development of the study.

4. Expand the conclusions taking into account the additions to the results and discussion sections (which are indicated above).

Best regards, the reviewer

Author Response

Dear authors, thank you for the opportunity to get acquainted with your interesting and unique developments. Your article has a number of advantages:

The introduction substantiates in detail the relevance of the study and the scientific study of the issue. The design of the study, the very procedure for developing the program is spelled out in great detail, both the principles of development, which allow assessing the scientific validity of the author's approach, and the content of the program, are covered.

The results are described in detail in terms of describing the content of the program, its structure and visual presentation of materials for managers.

Thank you for this positive feedback.

At the same time, it is necessary to make adjustments to some sections of your article for better understanding: The Usability testing section needs to be supplemented with more complete information about the sample of pilot testing: age, seniority of a managerial position, professional area, how many employees he manages, etc.

We have included all data that we collected (age, gender, managerial experience, and number of people they manage).

It is also necessary to include information on what questions were asked to the participants of the pilot study regarding the convenience of the program, their assessment of the acquired knowledge and etc.

We have now included all data from the pilot study, which includes items mapped to the Technology Acceptance Model (TAM).

 In the results section, it is necessary to add information in%, how many people from the pilot sample, which gave answers to questions about the convenience and usefulness of this program. And not only in a general way to present the main conclusions (as indicated in lines 203-209). Because until there is a reliable understanding of the effectiveness of the use of the program during the pilot phase and it is not indicated what specific changes were made to the program after the results of the pilot project. We need both statistics and qualitative conclusions. So far, the conclusions and discussion of the results are more of a theoretical nature.

We have added all data that we have available for the pilot and usability testing. This includes a table of results, and qualitative data from free text responses. The findings are reflected in the discussion and conclusions.

  1. It is necessary to single out in a separate section the limitations of the study and further directions for the development of the study.

Limitations of the study has been added to the discussion:

“Study strengths and limitations

A strength of the study was the rigorous process for content and technical development, involving participatory design with multi-method approaches to co-creation, adapted in line with guidance for conducting participatory research during the COVID-19 pandemic. Our intervention was based on a Theory of Change and user-centered active learning pedagogy, and the research was theoretically informed, using Chisholm’s co-design model [44], and TAM [40], an information systems theory. The resource has been pilot tested across sectors (public, private and third) and organization types and sizes, which has allowed us to explore the usability and fidelity of the training program prior to full feasibility testing. We did not collect demographic data for the individuals who participated in the expert panel, to maintain confidentiality. In our empirical pilot study, the sample size was appropriate for usability testing, although not all managers were able to complete all the modules, due to the pressures and additional work demands related to the COVID-19 pandemic. Nevertheless, between 5-11 managers provided feedback on each module and this was deemed sufficient to meet the aims of the pilot testing and usability study. Managers involved in our expert panel and usability study were primarily from office-based roles, and the training program is yet to be feasibility tested in diverse employment settings.

The further directions for development are already presented in the conclusions – the next step is to test the feasibility and acceptability of the intervention in diverse employment settings. “

This has also been referred to throughout the manuscript:

  • Our abstract aim: “The aim of the study was to describe the design and development of the Managing Minds at Work digital training programme, prior to feasibility testing”

  • Our abstract conclusion: “The next step is to explore the feasibility and acceptability of this intervention with line managers in diverse employment settings”.

  • Our introduction, rationale: “The research will be conducted over several phases, including intervention development, feasibility testing, evaluation, and exploration of implementation processes. In this paper, we report the first stage of the research, which is the intervention development”.

  • Our methods: “Managing Minds at Work was then hosted on a Web platform in September 2021 ready for feasibility testing, a future phase of the research”.

  • Our methods: “Strategies to maintain or enhance fidelity will be explored in a future feasibility trial”.

  • Our discussion: “…we aim to consider the impact of the circumstances of the pandemic for the intervention development and future feasibility testing” and “the training programme is yet to be tested in diverse employment settings which is the next stage of the research”.

  • Our conclusions: “The next step is to explore the feasibility and acceptability of the Managing Minds at Work intervention with line managers in diverse employment settings, explore mechanisms for most effective delivery, and examine whether and how the intervention can be tailored for different types of organisations, and integrated alongside existing polices, practices and interventions:”.

  1. Expand the conclusions taking into account the additions to the results and discussion sections (which are indicated above).

We have re-drafted the discussion and conclusions to reflect the new findings added (pilot and usability testing data).

Reviewer 5 Report

I am delighted to have reviewed this article and I congratulate you on the quality and style of the paper. It was both interesting and pleasurable to review.

I have a few very minor suggestions that I hope help improve general readership.

Figures 2,3,5 and 6 were slightly difficult to read - exploring size, font, colour may help.

It would be nice to understand the basic socio demographic characteristics of the expert panel, and participants to explore the  how gender, culture, ethnicity, age were accounted for in the design and acceptability study.

First paragraph of the discussion seems a little repetitive, consider revising and shortening.

The principles in the discussion bring in new concepts which is contrary to convention - these may be better suited in methods.

A small statement on strengths and limitations is required in your discussion.

Again great work and I really look forward to watching the broader implementation of this work.

Author Response

I am delighted to have reviewed this article and I congratulate you on the quality and style of the paper. It was both interesting and pleasurable to review.

Thank you for this positive feedback.

I have a few very minor suggestions that I hope help improve general readership.

Figures 2,3,5 and 6 were slightly difficult to read - exploring size, font, colour may help.

We have altered the colour scheme on the figures that we have created for this paper (to a lighter shade to reduce the contrast) and we hope this has improved readability. The other figures are taken directly from the training programme as they currently appear so these can’t be changed at this point, or they would not represent the developed programme being tested for usability here. The figure showing the summary of the training materials needs to appear on a single page, although we did try various versions, our stakeholders agreed it looked better on a single page and readers should be able to expand this to make the text more readable on their own devices if they wish.

However, we will also take these important comments on board for our future research and development of these materials.

It would be nice to understand the basic socio demographic characteristics of the expert panel, and participants to explore how gender, culture, ethnicity, age were accounted for in the design and acceptability study.

We have described the expertise that existing within the expert panel and how this aligned with the expertise of the research team:

“A virtual panel was established in May 2021 consisting of 10 expert stakeholders who contributed to the co-design of training content. As members of the virtual panel, stakeholders liaised with the project team, through a range of methods including individual email correspondence and/or telephone discussion, and/or group discussion. Stakeholders included individuals from academic institutions, local authorities, the mental health charity Mind and Public Health England. The stakeholder group had expertise in education and training, specialists in human resources, employment and/or mental health), lived experience of mental ill-health and line management, including supervision or management of employees with mental health concerns. The research team had expertise in occupational and health psychology, mental health, workforce and employment issues, digital health education, and equality, diversity, and inclusion. Collaborative working between the research team and stakeholders allowed for evidence-based (top-down) and experience-based (bottom-up) input to design.”

We did not collect demographic data on expert panel members (for reasons of maintaining confidentiality) although in response to your comment, this has been added as a study limitation.

However, we did collect demographic characteristics of the managers who participated in the pilot study, and this has now been included.

First paragraph of the discussion seems a little repetitive, consider revising and shortening.

We have completely re-drafted the discussion.

The principles in the discussion bring in new concepts which is contrary to convention - these may be better suited in methods.

Thank you for pointing this out – we have moved the guidance for conducting participatory research during a pandemic to the methods section. The discussion has been re-drafted and is brief to adhere to its purpose as a reflection on the development and usability testing of a digital intervention.

A small statement on strengths and limitations is required in your discussion.

This has been added:

“Study strengths and limitations

A strength of the study was the rigorous process for content and technical development, involving participatory design with multi-method approaches to co-creation, adapted in line with guidance for conducting participatory research during the COVID-19 pandemic. Our intervention was based on a Theory of Change and user-centered active learning pedagogy, and the research was theoretically informed, using Chisholm’s co-design model [44], and TAM [40], an information systems theory. The resource has been pilot tested across sectors (public, private and third) and organization types and sizes, which has allowed us to explore the usability and fidelity of the training program prior to full feasibility testing. We did not collect demographic data for the individuals who participated in the expert panel, to maintain confidentiality. In our empirical pilot study, the sample size was appropriate for usability testing, although not all managers were able to complete all the modules, due to the pressures and additional work demands related to the COVID-19 pandemic. Nevertheless, between 5-11 managers provided feedback on each module and this was deemed sufficient to meet the aims of the pilot testing and usability study. Managers involved in our expert panel and usability study were primarily from office-based roles, and the training program is yet to be feasibility tested in diverse employment settings.”

Again, great work and I really look forward to watching the broader implementation of this work.

Thank you for this positive feedback.

Round 2

Reviewer 2 Report

Dear authors,

The improvements of the paper is visible. The theoretical background and methodological approach is better supported. Inserting the usability testing and associated TAM construct table provide more clear insight about the sample of the research. Also insertion of responses of the managers brings a qualitative plus.

To topic is original, the paper in actual form covers a research gap and provides a solid theoretical, qualitative, quantitative and practical insight for mental health management at work.

Reviewer 3 Report

The article has already been rejected!!!

Reviewer 4 Report

Dear authors, thank you for taking into account all the recommendations. In the latest version, your additions to the pilot study provide information about the effectiveness of the program and the necessary information for its improvement.

Best regards, the reviewer